# Bladder Dysfunction in Sickle Cell Disease Is Associated with Inflammation and Oxidative Stress

**DOI:** 10.3390/ijms26199776

**Published:** 2025-10-08

**Authors:** Dalila Andrade Pereira, Fabiano Beraldi Calmasini, Tammyris Helena Rebecchi Silveira, Danillo Andrade Pereira, Mariana G. de Oliveira, Fernando Ferreira Costa, Fábio Henrique Silva

**Affiliations:** 1Laboratory of Pharmacology, São Francisco University Medical School, Bragança Paulista 12916-900, SP, Brazil; dalilaandpereira@gmail.com (D.A.P.); tammyhelena97@hotmail.com (T.H.R.S.); danilloandrade83@gmail.com (D.A.P.); mariana.taranto@usf.edu.br (M.G.d.O.); 2Department of Pharmacology, Paulista School of Medicine, Federal University of São Paulo, São Paulo 04044-010, SP, Brazil; fabiano.calmasini@unifesp.br; 3Hematology and Hemotherapy Center, University of Campinas, Campinas 13000-000, SP, Brazil; ferreira@unicamp.br

**Keywords:** detrusor, IL-1β, myeloperoxidase, NADPH oxidase, overactive bladder

## Abstract

Bladder dysfunction, particularly overactive bladder (OAB), is increasingly recognized as a clinical concern in patients with sickle cell disease (SCD), yet its pathophysiological mechanisms remain poorly understood. This study investigated the relationship between oxidative stress, inflammation, and bladder dysfunction in the Townes transgenic SCD mouse model. Cystometric analysis revealed that SCD mice exhibit an OAB phenotype, characterized by increased frequencies of voiding and non-voiding contractions and reduced bladder compliance. In vitro functional assays demonstrated detrusor hypocontractility in SCD mice, associated with a significant reduction in carbachol- and EFS-induced contractions and downregulation of muscarinic M3 receptor expression. Purinergic signaling and calcium-dependent contractility remained preserved. Molecular analyses showed increased mRNA expression of NOX-2 and IL-1β, and elevated protein levels of 3-nitrotyrosine and myeloperoxidase (MPO) activity, indicating redox imbalance and chronic inflammation in bladder tissue. Together, these changes suggest that oxidative and nitrosative stress, combined with inflammation, contribute to bladder remodeling and dysfunction in SCD. This is the first study to characterize bladder alterations in Townes SCD mice, establishing this model as a valuable tool for investigating lower urinary tract complications in SCD. Our findings provide mechanistic insight into the genitourinary manifestations of SCD and identify redox and inflammatory pathways as potential therapeutic targets for bladder dysfunction in affected individuals.

## 1. Introduction

Sickle cell disease (SCD) is an autosomal recessive genetic disorder caused by a single amino acid substitution in the β-globin chain, leading to the production of abnormal hemoglobin S (HbS) [1]. Under hypoxic or dehydrated conditions, HbS polymerizes, causing red blood cells to assume a sickle shape. These sickle-shaped erythrocytes are less flexible and have a shorter lifespan, resulting in intravascular and extravascular hemolysis. These hemolytic events, along with the accompanying chronic inflammation, oxidative stress, and vaso-occlusion, are key drivers of the diverse clinical manifestations and complications of SCD, including various forms of organ dysfunction [1].

Among these complications, bladder dysfunction, particularly overactive bladder (OAB), is increasingly recognized as a significant clinical concern for SCD patients. While the prevalence of OAB in the general population of the United States ranges from 16.5% to 23.2%, it is estimated to be as high as 40% among individuals with SCD [2,3,4]. Symptoms of OAB in SCD patients include urinary frequency, urgency, and nocturia [2,3]. The higher prevalence of OAB in SCD patients suggests that this population is particularly vulnerable to bladder dysfunction, which can significantly impair their quality of life.

The lower urinary tract functions of storing and releasing urine are regulated by neural circuits in the brain, spinal cord, and peripheral ganglia. The contraction of the detrusor smooth muscle in the bladder is primarily mediated by the activation of M3 muscarinic receptors through the cholinergic action of acetylcholine (ACh), which is released from parasympathetic nerve fibers [5,6]. M2 receptors are also expressed in the detrusor and mainly support bladder contraction by inhibiting relaxation mechanisms [5,7]. Both M2 and M3 receptor activation lead to increased intracellular Ca^2+^ levels and subsequent muscle contraction [5,6,8]. While ACh is the primary neurotransmitter released from parasympathetic nerve terminals, the parasympathetic innervation also releases adenosine triphosphate (ATP). ATP induces detrusor contraction by activating purinergic receptors, such as P2X1, further increasing intracellular Ca^2+^ levels [6]. Alterations in the expression of these receptors (M2, M3, and P2X1) can contribute to bladder dysfunction; however, their expression has not yet been determined in the bladder within the context of SCD.

Bladder dysfunction has been extensively studied under various experimental conditions. However, only a few studies have specifically investigated the pathophysiological changes associated with SCD [9,10,11,12]. Research studies using SCD mice have identified voiding dysfunction characterized by increased voiding frequency and non-voiding contractions associated with detrusor hypercontractility [10,12]. This study highlighted the role of nitric oxide (NO) signaling dysregulation in mediating these effects, suggesting that deficiencies in NO could contribute to lower urinary tract symptoms in SCD [10,12]. Additional research by Karakus et al. (2020) further explored the molecular pathways involved, demonstrating that the dysregulation of the RhoA/Rho-associated kinase (ROCK) pathways is also linked to detrusor overactivity in SCD mice [11]. In contrast, another study by Claudino et al. (2015) described an underactive bladder phenotype in older SCD mice, suggesting a potential progression from overactive bladder to underactive bladder with aging [9]. However, this progression has not been confirmed in human studies, indicating the need for further investigation to clarify these mechanisms in human patients.

Although studies have demonstrated that oxidative stress, driven by increased superoxide production from the NOX-2 isoform of NADPH oxidase, contributes to the pathophysiology of OAB in various animal models, nitrosative stress, resulting from the formation of reactive nitrogen species, such as peroxynitrite, also plays a significant role [13,14,15,16,17]. Additionally, a mouse model with phenylhydrazine (PHZ)-induced hemolysis exhibits an OAB phenotype associated with elevated oxidative stress in the bladder [17]. Chronic inflammation has also been implicated in the development of OAB in both human and animal models [18,19,20,21,22,23]. However, no study has explored the interplay between inflammatory factors, oxidative stress, and OAB, specifically in SCD. Thus, it is critical to elucidate how these factors contribute to bladder dysfunction in SCD.

Transgenic SCD mice have been essential in advancing our understanding of the pathophysiology of SCD [24]. Studies investigating bladder dysfunction in SCD have only used the transgenic Berkeley SCD mice model, which exclusively expresses human sickle hemoglobin and closely replicates the main characteristics observed in human SCD [9,10,11,12,25]. Another transgenic mouse model, the Townes SCD mouse, was developed by replacing the murine α-globin gene with the human α-globin gene and the murine β-globin gene with the human γ and βS globin genes [24]. However, bladder alterations in the Townes model remain unexplored.

Considering the lack of previous studies exploring the direct relationship between oxidative stress, inflammation, and functional bladder alterations in the Townes sickle cell disease model, this study aimed to investigate these associations for the first time in this animal model. We hypothesized that oxidative stress and chronic inflammation, well-established features of sickle cell disease, are key contributors to the development of OAB clinically observed in patients. To test this hypothesis, we assessed critical markers of oxidative stress (NOX-2, 3-nitrotyrosine [3-NT]), inflammation (myeloperoxidase [MPO], interleukin-1 beta [IL-1β]), and the expression of muscarinic (M2, M3) and purinergic (P2X1) receptors. The methodological innovation of this study lies precisely in the pioneering use of the Townes model to investigate bladder dysfunction, providing new perspectives for identifying specific and translational therapeutic targets.

## 2. Results

### 2.1. In Vivo Evaluation of Bladder Function

Figure 1 presents representative cystometric traces from control and SCD mice. In control animals, cystometry revealed regular micturition cycles with infrequent non-voiding contractions (NVCs) (Figure 1A). In contrast, SCD mice exhibited an irregular micturition pattern (Figure 1B), accompanied by a significant increase in both voiding frequency and the number of NVCs compared to controls (*p* < 0.05; Figure 2A,B). Furthermore, bladder compliance was significantly reduced in SCD mice (*p* < 0.05; Figure 2C). No significant differences were observed in peak pressure, threshold pressure, or bladder capacity between control and SCD groups (Figure 2D–F).

### 2.2. Functional Analysis of Detrusor Smooth Muscle Contractility

EFS (2–32 Hz) elicited frequency-dependent contractions of detrusor smooth muscle (DSM) in both groups, although the responses were consistently lower in SCD mice at all frequencies tested (Figure 3A). The muscarinic receptor agonist carbachol (1 nM–10 µM) induced concentration-dependent contractions in isolated DSM strips from both control and SCD mice (Figure 3B). Maximal responses (E_max_) were significantly reduced in DSM from SCD mice compared to controls (*p* < 0.01), with values of 1.39 ± 0.40 and 2.84 ± 0.17 mN/mg, respectively (*n* = 5). However, no significant differences were observed in the pEC_50_ values for carbachol between the control and SCD groups (6.89 ± 0.04 and 6.87 ± 0.17, respectively).

In separate experiments, α,β-methylene ATP (1 and 3 µM) produced DSM contractions that were similar between control and SCD groups (Figure 3C; *n* = 4). To evaluate receptor-independent contractility, cumulative addition of KCl induced concentration-dependent DSM contractions in both groups (Figure 3D). No significant differences in Emax or pEC_50_ values for KCl were found between control (3.84 ± 0.81 mN/mg and 1.30 ± 0.12, respectively; *n* = 4) and SCD mice (4.29 ± 0.49 mN/mg and 1.30 ± 0.08, respectively; *n* = 4).

The average weight of the detrusor strips used for functional recordings did not differ significantly between control and SCD mice (4.8 ± 0.49 mg vs. 4.45 ± 0.27 mg, respectively; n = 16), supporting the validity of contractility normalization by tissue weight.

### 2.3. mRNA Expression of M2, M3, and P2X1 Receptors

Muscarinic (M2 and M3) and purinergic (P2X1) receptors play critical roles in bladder function by mediating detrusor smooth muscle contraction and regulating overall bladder dynamics. The mRNA expression of muscarinic M3 receptors in the detrusor was 55.17% lower in SCD mice compared to controls (*p* < 0.05; Figure 4B). In contrast, no significant differences were observed in the mRNA expression levels of muscarinic M2 receptors (Figure 4A) or purinergic P2X1 receptors (Figure 4C) between the two groups.

### 2.4. mRNA Expression of NOX-2 and SOD-1, and Protein Expression of 3-Nitrotyrosine

NOX-2 is a key enzyme involved in the production of reactive oxygen species (ROS), thereby contributing to oxidative stress, whereas SOD-1 functions as an antioxidant enzyme that neutralizes ROS [26]. 3-NT is a marker of nitrosative stress, formed when reactive nitrogen species, such as peroxynitrite, react with tyrosine residues on proteins [27]. In SCD mice, the mRNA expression of NOX-2 in the bladder was approximately twofold higher than in control animals (*p* < 0.05; Figure 4D), indicating increased ROS production and oxidative stress in the bladder tissue. No significant differences were observed in the mRNA expression of SOD-1 between the groups (Figure 4E). In addition, the protein expression of 3-NT in bladder tissue was approximately twofold higher in SCD mice compared to controls (*p* < 0.05; Figure 5), reflecting enhanced nitrosative stress.

### 2.5. mRNA Expression of IL-1β

Interleukin-1 beta (IL-1β) is a pro-inflammatory cytokine that plays a pivotal role in initiating and sustaining inflammatory responses [28]. In SCD mice, the mRNA expression of IL-1β in bladder tissue was approximately tenfold higher compared to control animals (*p* < 0.05; Figure 4F). This marked increase suggests a heightened inflammatory state in the bladder of SCD mice, which may contribute to or exacerbate bladder dysfunction and overactive bladder symptoms by promoting local inflammation, tissue damage, and altered smooth muscle contractility.

### 2.6. Myeloperoxidase Activity

MPO plays an important role in the initiation and progression of acute and chronic inflammatory diseases. MPO activity was significantly higher (*p* < 0.01) in bladders from SCD mice compared to the control (Figure 6), indicating an elevated inflammatory response and the presence of neutrophil infiltration in the bladder tissue.

### 2.7. Bladder Characteristics and Histopathological Analysis

No significant differences in body weight were observed between SCD and control mice (Table 1). However, both absolute bladder weight and the bladder-to-body weight ratio were significantly higher in SCD mice (*p* < 0.05), suggesting the presence of inflammation and/or tissue remodeling. To evaluate structural changes in bladder tissue, histological analysis was performed on hematoxylin and eosin (H&E)-stained bladder sections. 

Bladders from control animals (Figure 7A–C) exhibited a well-organized bladder wall with an intact urothelium. Although bladders from SCD mice maintained the general structural organization, they exhibited marked inflammatory cell infiltration, predominantly by mononuclear cells, in the mucosal layer (Figure 7D–F). Inflammatory infiltration, predominantly by mononuclear cells, was observed in the submucosal and muscular layers of the bladder in SCD mice.

Morphometric analysis (Table 1) revealed a 20% increase in detrusor smooth muscle thickness in SCD mice compared to controls (335 ± 15.7 vs. 283 ± 13.3 µm, respectively; *p* < 0.05), indicating a hypertrophic response of the detrusor muscle. In addition, a significant reduction in urothelial thickness was observed in SCD mice (*p* < 0.05), suggesting potential urothelial damage or atrophy, which may compromise bladder barrier function and/or sensory signaling.

## 3. Discussion

In this study, Townes SCD mice exhibit a pronounced overactive bladder phenotype, characterized by irregular micturition patterns and a significant increase in voiding and non-voiding contractions. This bladder dysfunction is associated with the upregulation of NOX2 expression, reduced expression of M3 muscarinic receptors, and detrusor hypocontractility. Additionally, the bladder tissue of these transgenic SCD mice shows signs of inflammation, such as cellular infiltration and increased MPO activity. These findings align with clinical manifestations frequently reported in SCD patients, who commonly experience bladder dysfunction and increased urinary frequency. The similarities between the bladder dysfunction in Townes SCD mice and the symptoms observed in human patients underscore the translational relevance of this model, providing insight into the pathophysiological mechanisms underlying bladder dysfunction in SCD.

The cystometric findings in this study revealed a significant increase in both voiding and non-voiding contractions in Townes SCD mice, indicating an irregular micturition pattern consistent with an OAB. These results align with clinical studies that have reported symptoms of OAB in patients with SCD [2,3] and with findings in young Berkeley transgenic mice, which also demonstrated increased frequencies of voiding and non-voiding contractions [10]. In contrast, Claudino et al. reported an underactive bladder phenotype in older SCD mice, suggesting that overactive bladder in SCD may progress to underactive bladder over time [9].

Parasympathetic activity, mediated by the release of ACh, induces detrusor muscle contraction primarily through the activation of M2 and M3 muscarinic receptors, with the M3 receptors being more functionally significant for detrusor contractions [5,6]. In our in vitro functional assays, we observed a significant reduction in neurogenic contractions induced by EFS in SCD mice. To determine if this reduction was specifically related to M3 muscarinic receptor activation, we further evaluated detrusor contractions induced by CCh, a specific M3 muscarinic receptor agonist. The contractions induced by CCh were also significantly lower in SCD mice than in controls, suggesting a potential downregulation or desensitization of M3 muscarinic receptors. This hypothesis is further supported by the mRNA expression data, which demonstrated a decreased expression of M3 receptors in the bladder tissue of SCD mice. Similarly, bladders from patients with bladder outlet obstruction and detrusor overactivity/hypersensitivity have also been shown to display lower expression of M3 muscarinic receptors [20].

Additionally, we assessed the role of purinergic signaling in detrusor contractions by using α,β-methylene ATP, an agonist for P2X1 receptors, which produced concentration-dependent DSM contractions [29]. In contrast, responses to α,β-methylene ATP, a P2X1 receptor agonist, were unchanged, indicating preserved purinergic signaling. Similarly, no significant differences were observed in detrusor contractions induced by KCl, which causes receptor-independent contractions by elevating intracellular Ca^2+^, suggesting that calcium channels are unaffected in SCD mice. These findings suggest that reduced contractility in Townes SCD mice is due to the downregulation of M3 receptors rather than intrinsic muscle defects.

Comparatively, young Berkeley mice displayed increased detrusor contractile responses to EFS, carbachol, and KCl [10], indicating a possible early-stage OAB phenotype, while older Berkeley mice exhibited reduced detrusor contractions [9], suggesting a progression to an underactive bladder state over time. Depending on age or genetic background, these differences highlight the potential progression and variability of bladder dysfunction in SCD. This contrast with the Townes SCD mice, which consistently demonstrated reduced contractility, underscores the complexity of SCD-associated bladder dysfunction and suggests that OAB in SCD may initially present as an OAB phenotype in younger individuals, potentially progressing to an underactive phenotype with age. Understanding these variations is crucial for developing targeted therapeutic strategies for the specific stages of bladder dysfunction in SCD patients.

MPO is an enzyme with antimicrobial functions released by activated polymorphonuclear neutrophils, playing a critical role in generating reactive oxygen species (ROS), which are potent mediators of tissue damage and oxidative stress [30,31]. In the broader context of SCD, chronic inflammation is a well-documented feature, with SCD patients and Berkeley sickle cell mice showing elevated plasma MPO levels [31,32]. Similarly, isolated thoracic aortas from SCD mice also exhibit high levels of MPO [31].In our study, increased MPO activity in the bladders of Townes SCD mice suggests a heightened inflammatory response, likely due to neutrophil infiltration. This elevated MPO activity and subsequent ROS production likely contribute to oxidative stress, tissue damage, and impaired bladder function. Additionally, our findings showed a significant increase in the gene expression of IL-1 beta, a key pro-inflammatory cytokine, in the bladder tissue of SCD mice. This inflammatory environment can promote local tissue damage, contributing to or exacerbating bladder dysfunction and OAB symptoms. The upregulation of IL-1 beta and MPO activity in the bladder has been associated with overactivity in animal models [18,19,33]. Supporting these observations, Wang and Kuo identified markers of chronic inflammation in bladder biopsies from patients with OAB, both with and without diabetes, suggesting that inflammation can trigger a hypertrophic bladder phenotype regardless of its origin [21]. While the pathophysiology of OAB is complex and not fully understood, it likely involves multiple mechanisms, including myogenic, neurogenic, inflammatory, and idiopathic causes [34,35,36]. Our findings emphasize the role of inflammation, particularly the increased activity of MPO and expression of IL-1 beta, as significant factors contributing to bladder dysfunction in SCD.

Increased oxidative stress, characterized by elevated ROS production or reduced antioxidant capacity, is associated with the development of OAB in experimental models and is an important factor in the pathophysiology of SCD [14,16,37,38]. NOX-2, an important NADPH oxidase isoform, catalyzes electron transfer to oxygen, generating superoxide anions [39], while SOD-1 is a crucial antioxidant enzyme that catalyzes the dismutation of superoxide anions into hydrogen peroxide and oxygen, thereby playing a protective role against oxidative damage. Our findings showed no changes in the mRNA expression for SOD-1, whereas we observed increased NOX-2 expression in the detrusor muscle of Townes SCD mice, suggesting elevated O2^−^ production in the bladder. The excess O2^−^ can react with NO to form peroxynitrite (ONOO^−^), a highly reactive nitrogen species [40]. ONOO^−^ can react with tyrosine residues to produce 3-NT, a marker of nitrosative stress and tissue damage [27]. Our study found increased 3-NT expression in the detrusor muscle of Townes SCD mice, indicating that enhanced nitrosative damage may contribute to bladder dysfunction in SCD. Similarly, elevated NOX-2 and 3-NT expression in the penis is associated with the pathophysiology of priapism in mice and patients with SCD [37,41]. This oxidative stress is a driver of molecular dysfunction and manifests in significant structural changes within the bladder tissue [42].

Our study shows significant structural alterations in the bladders of SCD mice, characterized by increased bladder wall thickness, which correlates with the observed increase in bladder weight. These findings suggest a remodeling process within the bladder wall, likely due to chronic functional demands, such as the increased voiding and non-voiding contractions typically seen in OAB conditions. The decreased urothelial thickness in SCD mice may also indicate urothelial damage or atrophy. Bladder control by the central and autonomic nervous systems requires adequate mechanosensory input from the urothelium, which is regulated by various receptors and ion channels, mechanoreceptors, and transmitters or mediators such as neurotrophins, peptides, ATP, acetylcholine, prostaglandins, prostacyclin, and NO [43]. Therefore, urothelial dysfunction in SCD mice may compromise normal bladder function, contributing to bladder overactivity.

Similarly to the bladder remodeling seen in humans with bladder outlet obstruction and in animal models of obstruction, which includes cellular hypertrophy and changes across multiple bladder compartments (e.g., urothelium, suburothelium, detrusor smooth muscle cells, extracellular matrix, and neurons) [44,45], the changes observed in SCD mice may be driven by analogous mechanisms. Cyclic stretch, increased pressure, and hypoxia have been shown to modulate multiple signaling pathways involved in these processes [45].

Histological evaluation further showed inflammatory cell infiltration, predominantly mononuclear cells, in both the submucosa and muscular layers of the bladder, along with signs of edema in the bladder tissue of SCD mice. The infiltration of inflammatory cells suggests ongoing inflammation, potentially leading to tissue damage, further reducing bladder compliance, and contributing to overactivity. The presence of edema, characterized by fluid accumulation within the bladder tissue, indicates increased vascular permeability or damage, possibly due to inflammatory processes.

## 4. Material and Methods

### 4.1. Animals

All animal procedures and experimental protocols were conducted in accordance with the Ethical Principles in Animal Research adopted by the Brazilian College for Animal Experimentation (COBEA) and followed the guidelines of the Guide for the Care and Use of Laboratory Animals. The experimental protocols were approved by the Ethics Committee on the Use of Animals (CEUA) at the University of Campinas (protocol number 3909–1). Male C57BL/6 mice (controls) and male Townes transgenic sickle cell mice, aged 3 to 4 months, were used in this study. Breeding pairs of Townes mice were originally developed at The Jackson Laboratory (Bar Harbor, ME, USA) and acquired by the Multidisciplinary Center for the Investigation of Biological Sciences in Laboratory Animals (CEMIB), University of Campinas (UNICAMP, Campinas, Sao Paulo, Brazil). These mice were bred locally at CEMIB, and offspring were genotyped to identify homozygous animals for SCD. Experimentally eligible animals were then purchased from CEMIB and transferred to the animal facility of the Laboratory of Molecular Biology and Hemostasis at the Hematology and Hemotherapy Center (Hemocentro-UNICAMP), where all experimental procedures were performed. Mice were housed in groups of three per cage under a 12 h light–dark cycle, with food and water available ad libitum.

### 4.2. Cystometry (In Vivo)

Mice were anesthetized with an intraperitoneal injection of urethane (1.8 g/kg). A 1 cm longitudinal incision was made in the abdominal wall to expose the bladder. A 25G butterfly needle was inserted into the bladder body and connected to a pressure transducer and an infusion pump. Prior to initiating cystometry, the bladder was completely emptied. Cystometric recordings were performed by continuously infusing saline into the bladder at a flow rate of 0.6 mL/h.

The following parameters were assessed during the recordings: bladder capacity, calculated as the time to the first micturition multiplied by 0.6 (mL) and divided by 60 (min); threshold pressure, defined as the pressure recorded immediately before the onset of micturition; compliance, determined as the ratio between capacity and threshold pressure (mL/cmH_2_O); peak pressure, defined as the pressure at the onset of micturition; and the frequency of voiding contractions, expressed as the number of micturition cycles per minute. In addition, non-voiding contractions (NVCs) were identified as spontaneous increases in intravesical pressure greater than 4 mmHg above baseline that were not associated with urine expulsion. NVCs were not accompanied by voiding events.

### 4.3. Functional Studies in Detrusor Smooth Muscle Strips and Concentration–Response Curves

Mice were anesthetized with isoflurane, and the urinary bladders were excised and sectioned horizontally at the level of the ureters. Two longitudinal detrusor smooth muscle (DSM) strips with intact urothelium were obtained from each bladder. The strips were mounted in 4 mL organ baths containing Krebs–Henseleit solution (117 mM NaCl, 4.7 mM KCl, 2.5 mM CaCl_2_, 1.2 mM MgSO_4_, 1.2 mM KH_2_PO_4_, 25 mM NaHCO_3_, and 11 mM glucose), continuously bubbled with a mixture of 95% O_2_ and 5% CO_2_ (pH 7.4) at 37 °C. Changes in isometric force were recorded using a PowerLab data acquisition system (LabChart software, version 7.0; AD Instruments, MA, USA). Resting tension was adjusted to 5 mN at the beginning of the experiments. The equilibration period was 60 min, and the bathing solution was replaced every 15 min.

Cumulative concentration–response curves to the full muscarinic agonist carbachol (1 nM to 10 μM) and to potassium chloride (KCl; 1 to 300 mM) were obtained in DSM strips. In separate experiments, electrical field stimulation (EFS)-induced contractions were evoked using the following parameters: 80 V, 1 ms pulse width, 10 s of stimulation, and 2 min intervals between pulses.

Nonlinear regression analysis to determine pEC_50_ values was performed using GraphPad Prism (GraphPad Software, Version 7, San Diego, CA, USA), with the constraint that Φ = 0. Concentration–response data were fitted to a logistic function in the following form:E = E_max_/([1 + (10^c^/10^x^)^n^] + Φ)
where *E* is the response at a given concentration of agonist, *E_max_* is the maximal response, *c* is the logarithm of the EC_50_ (the concentration producing 50% of the maximal effect), *x* is the logarithm of the agonist concentration, *n* is the Hill coefficient (slope factor), and Φ is the baseline response in the absence of agonist.

Values of pEC_50_ are presented as the mean ± SEM. Maximal response values (E_max_) were normalized to the wet weight of the corresponding DSM strips and expressed as mN per milligram of wet tissue.

### 4.4. Real-Time RT-PCR

Total RNA was extracted from mouse bladder samples using TRIzol Reagent (Invitrogen Corp., Carlsbad, CA, USA). Three-microgram RNA samples were incubated with 1 U DNase I (Invitrogen, Rockville, MD, USA) for 15 min at room temperature (RT). EDTA was added to a final concentration of 2 mM to stop the reaction. The DNase I enzyme was subsequently inactivated by incubation at 65 °C for 5 min. DNase I-treated RNA samples were reverse transcribed using SuperScript III and RNaseOUT (Invitrogen Corp., Carlsbad, CA, USA) for 50 min at 50 °C, followed by 15 min at 70 °C. cDNA concentrations were quantified using a NanoDrop spectrophotometer (ND-1000; NanoDrop Technologies Inc., Wilmington, DE, USA).

Primers for *CHRM2* (M2), *CHRM3* (M3), *P2RX1* (P2X1), *CYBB* (NOX-2), *SOD1* (SOD-1), *IL1B* (IL-1β), *GAPDH*, and *ACTB* (β-actin) were designed using the PrimerExpress™ software (Applied Biosystems, Foster City, CA, USA) (Table 2). The optimal concentration for each primer pair was determined, and amplification efficiency was calculated using the equation E = 10^−1/slope^ to confirm the accuracy and reproducibility of the reactions. Amplification specificity was verified by performing a dissociation curve protocol.

qRT-PCRs were performed in duplicate using 6 μL SYBR Green Master Mix (Applied Biosystems), 10 ng of cDNA, and optimal quantities of each primer in a final volume of 12 μL. Reactions were carried out in MicroAmp Optical 96-well plates (Applied Biosystems) using a 7500 Fast Real-Time PCR System (Applied Biosystems). Two technical replicates were included per plate for each sample, and each sample was independently run in duplicate. Gene expression data were normalized using the geNorm method, based on the geometric mean of two validated reference genes (*β-actin* and *GAPDH*) [46]. Results are expressed as relative mRNA expression levels.

### 4.5. Western Blotting

Protein separation from detrusor tissue homogenates was performed by electrophoresis on a 4–20% polyacrylamide gel containing 0.1% sodium dodecyl sulfate (SDS-PAGE). Proteins were then transferred to a polyvinylidene difluoride (PVDF) membrane using a wet transfer system. To reduce nonspecific binding, membranes were blocked for 1 h at room temperature with blocking buffer (5% non-fat dry milk in 10 mM Tris, 100 mM NaCl, and 0.02% Tween-20). Membranes were incubated overnight (15–16 h) at 4 °C with the primary antibody: monoclonal anti-3-nitrotyrosine (anti-3-NT; 1:1000; Abcam, catalog no. ab7048, Cambridge, MA, USA). After washing, membranes were incubated with a secondary antibody (rabbit anti-mouse IgG; Abcam, catalog no. ab6728). Immunoreactive bands were visualized using an enhanced chemiluminescence (ECL) detection kit. Quantified densitometry results were normalized to β-actin.

### 4.6. Histological Analysis

Control and SCD mice were weighed, euthanized with isoflurane, and the bladders were excised, weighed, and fixed with paraformaldehyde (4%) for 24 h. Bladders were dehydrated, cleared in xylene, embedded in paraffin, sliced in transversal sections (4 μm), and stained with hematoxylin and eosin (H&E) for histological examination. H&E images were used to determine bladder wall and urothelium thickness. Digital images were obtained using a digital camera (Leica DFC 300FX) connected to a bright field microscope (Leica DM500B). Identical imaging conditions, including illumination intensity and camera exposure time, were applied to all photographs, and a blank-field image was used to correct illumination. Five images of each section were acquired, and histomorphometric analysis was performed using ImageJ 1.53t (NIH).

### 4.7. Myeloperoxidase Activity Assay

Myeloperoxidase (MPO) activity was assessed to quantify the extent of neutrophil accumulation in bladder tissue, following a previously described method [30]. Excised bladders were frozen in liquid nitrogen, pulverized, and homogenized in 0.05 M sodium phosphate buffer (pH 7.4) containing 0.5% hexadecyltrimethylammonium bromide (HTAB; Sigma-Aldrich, St. Louis, MO, USA). The homogenates were sonicated for 30 seconds, subjected to two cycles of freezing and thawing, and then centrifuged at 10,000 rpm for 5 min at 4 °C. The resulting supernatant was used to determine MPO activity by spectrophotometric reaction with O-dianisidine hydrochloride (Sigma-Aldrich, St. Louis, MO, USA). Absorbance was measured at 460 nm, and the change in optical density (OD) over 5 min was recorded. Results were expressed as units of MPO per milligram of protein.

### 4.8. Drugs and Chemicals

α,β-Methylene ATP, carbachol, KCl, and urethane were purchased from Sigma-Aldrich (St. Louis, MO, USA). All reagents used were of analytical grade. Stock solutions were prepared in deionized water, aliquoted, and stored at −20 °C. Working dilutions were prepared immediately prior to use.

### 4.9. Statistical Analysis

The number of animals was determined to achieve a statistical power of 0.80 to 0.85, with a Type I error probability (α) of 0.05. Statistical analyses were performed using GraphPad Prism software (GraphPad Software Inc., San Diego, CA, USA). Data are presented as the mean ± standard error of the mean (SEM) of N experiments. Functional and cystometric data were analyzed using a two-tailed unpaired Student’s *t*-test. MPO activity, histological analysis, Western blot, and real-time RT-PCR data were analyzed using a two-tailed Mann–Whitney U test. A *p*-value < 0.05 was considered statistically significant.

## 5. Conclusions

This study demonstrates that Townes SCD mice exhibit a pronounced overactive bladder phenotype, characterized by irregular micturition patterns, increased frequencies of voiding and non-voiding contractions, and reduced bladder compliance. These functional abnormalities are associated with several molecular and structural changes, including the upregulation of NOX-2 expression, reduced expression of M3 muscarinic receptors, and increased markers of inflammation and oxidative stress, such as MPO, IL-1 beta, and 3-NT. Our findings also showed significant histological alterations, including increased bladder wall thickness, smooth muscle hypertrophy, urothelial thinning, and the presence of inflammatory infiltrates and edema, further supporting the role of sustained inflammation and oxidative stress in the pathophysiology of bladder dysfunction in SCD.

Collectively, these findings indicate that bladder dysfunction in SCD is linked to a complex interplay between inflammatory and oxidative mechanisms, promoting structural remodeling and impaired bladder function. Understanding these processes can lead to developing targeted therapeutic strategies to alleviate bladder symptoms in SCD patients, thereby improving their quality of life. Future studies are needed to explore potential interventions that may mitigate these pathophysiological changes and further elucidate the underlying mechanisms contributing to bladder dysfunction in SCD.

## Figures and Tables

**Figure 1 ijms-26-09776-f001:**
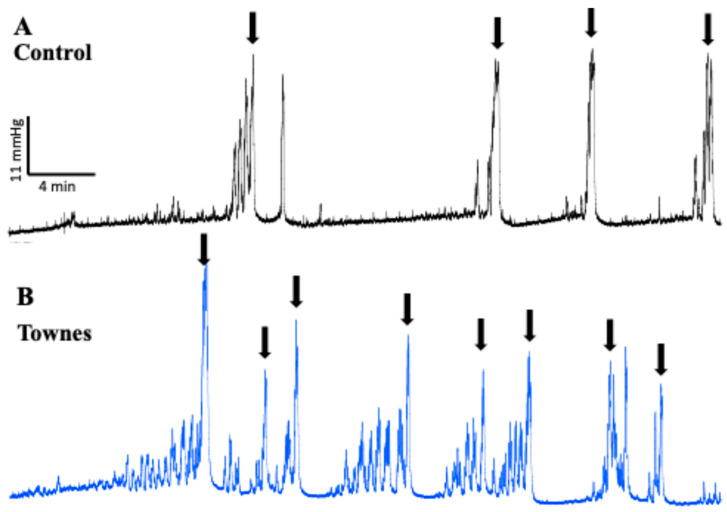
Representative cystometric tracings from (**A**) control and (**B**) SCD mice. Arrows in the cystometric trace indicate the micturition peaks.

**Figure 2 ijms-26-09776-f002:**
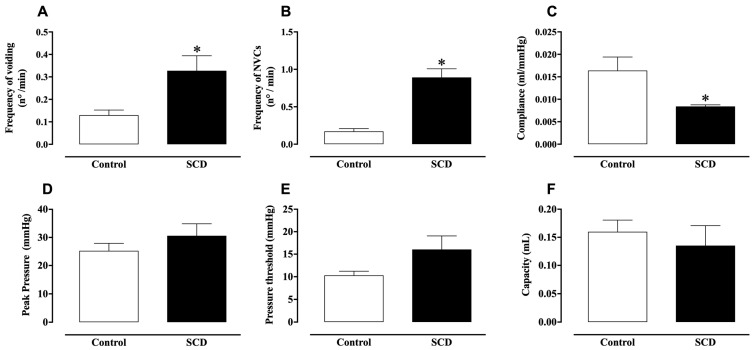
In vivo cystometric data of control and SCD mice. (**A**) Frequency of voiding, (**B**) frequency of NVCs, (**C**) compliance, (**D**) peak pressure, (**E**) pressure threshold, and (**F**) capacity. Data represent the means ± S.E.M. of 5–6 mice in each group. * *p* < 0.05 compared to the control group.

**Figure 3 ijms-26-09776-f003:**
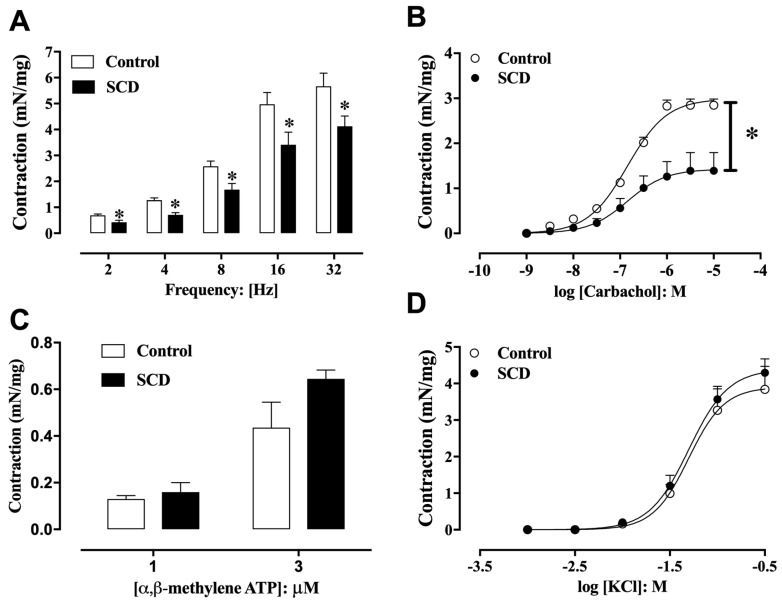
Concentration-dependent contraction curves induced by (**A**) EFS, (**B**) carbachol, (**C**) α,β-methylene ATP, and (**D**) KCl in the detrusor muscle of control and SCD mice. Data represent the means ± S.E.M. of 5–6 mice in each group. * *p* < 0.05 compared to the control group.

**Figure 4 ijms-26-09776-f004:**
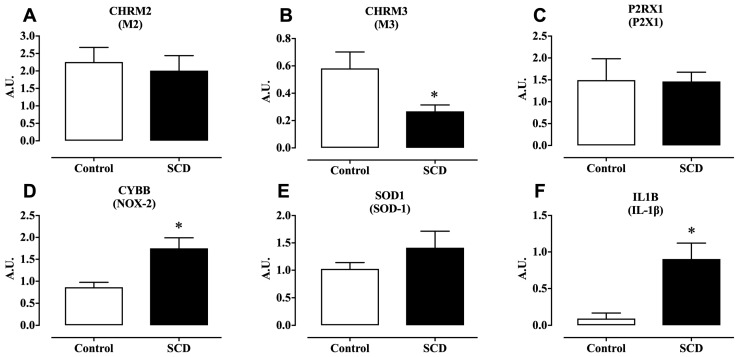
Gene expression (mRNA) in the detrusor of control and SCD mice. (**A**) M2. (**B**) M3. (**C**) P2X1. (**D**) NOX-2. (**E**) SOD-1. (**F**) IL-1 beta. Data represent the means ± S.E.M. of 5–6 mice in each group. * *p* < 0.05 compared to the control group. A.U.: arbitrary unit.

**Figure 5 ijms-26-09776-f005:**
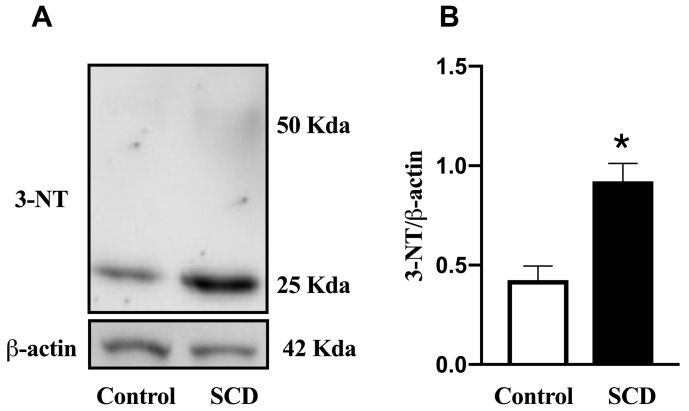
Protein expression for 3-NT in the bladder of control and SCD mice. (**A**) Representative Western blot. (**B**) Densitometric analysis of 3-NT normalized to β-actin, expressed as arbitrary units. Data represent the means ± S.E.M. of 4 mice in each group. * *p* < 0.05 compared to the control group.

**Figure 6 ijms-26-09776-f006:**
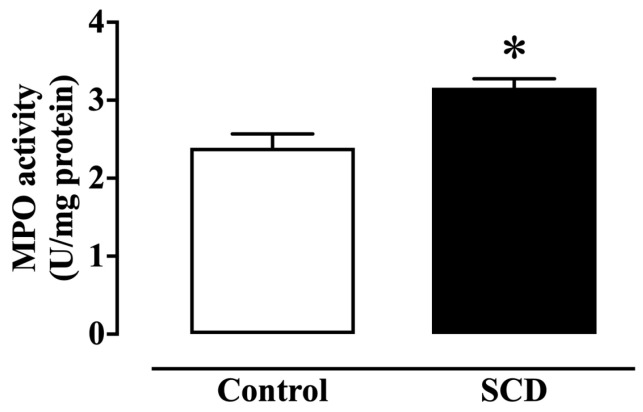
MPO activity in the detrusor of control and SCD mice. Data represent the means ± S.E.M. of 4 mice in each group. * *p* < 0.05 compared to the control group.

**Figure 7 ijms-26-09776-f007:**
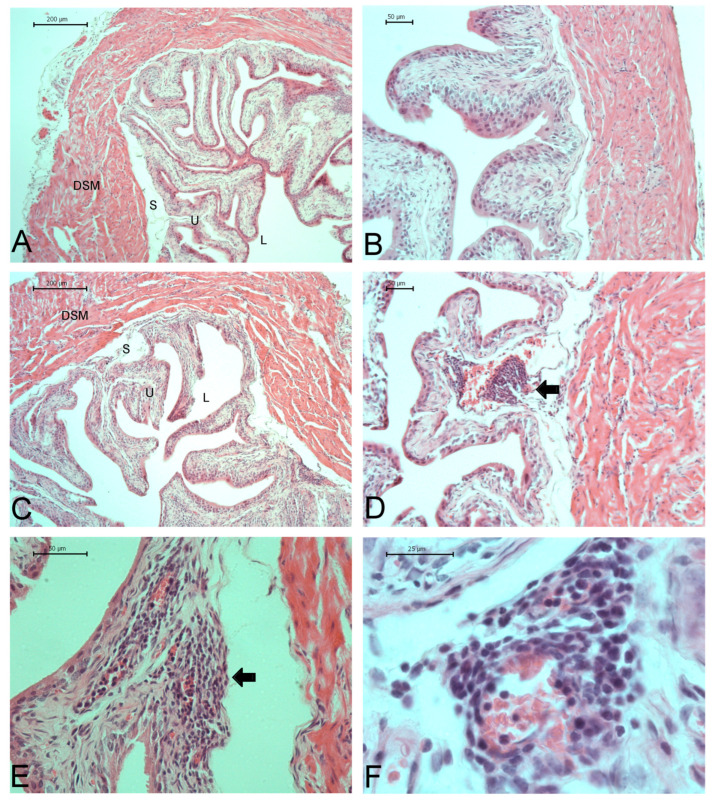
Histological sections of the bladder wall stained with hematoxylin and eosin from control (**A**–**C**) and SCD mice (**D**–**F**). DSM = detrusor smooth muscle; S = submucosa; U = urothelium; L = lumen. Black arrows = inflammatory cells. Scale bars: 200 μm (**A**,**C**), 50 μm (**B**,**D**), and 25 μm (**F**).

**Table 1 ijms-26-09776-t001:** Body weight, bladder weight, and histological analysis in control and SCD mice.

Parameters	Control	SCD
Body weight (g)	33.0 ± 1.06	31.59 ± 2.24
Bladder weight (mg)	29.98 ± 1.06	41.9 ± 1.32 *
Bladder weight: body weight	0.90 ± 0.02	1.34 ± 0.12 *
Bladder wall thickness (μm)	283.0 ± 13.36	335.2 ± 15.70 *
Urothelium thickness (μm)	45.93 ± 2.31	36.65 ± 0.51 **

Data represents the means ± SEM for four mice/group. * *p* < 0.05, ** *p* < 0.01 compared with the control group (two-tailed *t*-test).

**Table 2 ijms-26-09776-t002:** Sequences and ideal concentrations for the primers used in qRT-PCR.

Gene	Primer Sequence	Concentration
*CHRM2*–F	5′-ACACGGTTTCCCACTTCCCTG-3′	150 nM
*CHRM2*–R	5′-TGCATGCGTCACCCTTTTG-3′	
*CHRM3*–F	5′-CCCACAGGCAGTTCTCGAA-3′	150 nM
*CHRM3*–R	5′-CCTCCTAGATGACCGTTTCGT-3′	
*P2RX1*–F	5′-ATTCGCTTTGATATCCTTGTGG-3′	150 nM
*P2RX1*–R	5′-GCCGATGGTAGTCATAGTAGGG-3′	
*CYBB*–F	5′-TTGGGTCAGCACTGGCTCTG-3′	150 nM
*CYBB*–R	5′-TGGCGGTGTGCAGTGCTATC-3′	
*SOD1*–F	5′-CAGCATGGGTTCCACGTCCA-3′	150 nM
*SOD1*–R	5′-CACATTGGCCACACCGTCCT-3	
*IL1B*–F	5′-CAAGGAGACGGAATACAGGGC-3′	150 nM
*IL1B*–R	5′-CCAGGTCACCTCGACGTTTG-3	
*GAPDH*–F	5′-TGCACCACCAACTGCTTA-3′	70 nM
*GAPDH*–R	5′-GGATGCAGGGATGATGTTC-3′	
*ACTB*–F	5′-ACTGCCGCATCCTCTTCCT-3′	70 nM
*ACTB*–R	5′-GAACCGCTCGTTGCCAATA-3′	

F, forward; R, reverse.

## Data Availability

Data are available from the corresponding author upon reasonable request.

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
