# Peer review of "Bladder Dysfunction in Sickle Cell Disease Is Associated with Inflammation and Oxidative Stress"

_ijms, 2025, doi:10.3390/ijms26199776_

Round 1

Reviewer 1 Report

Comments and Suggestions for Authors

In this manuscript, the authors attempt to shed light on the etiology of bladder dysfunction in sickle cell disease (SCD) patients, which is becoming increasingly well known to be a morbidity associated with the disease.  Their research focuses on the relationship between SCD, oxidative stress, inflammation and bladder dysfunction.  They show data supporting increased oxidative stress in the bladder of Townes mouse model of SCD, which correlates with increased voiding frequency and decreased contractile force of detrusor smooth muscle (DSM). Moreover, they show decreased muscarinic receptor expression in the DSM and increased edema and pro-inflammatory cytokines in SCD mice. They conclude that the manifestations of bladder pathology in SCD patients is related to increased oxidative/nitrosative stress.

Major Concerns:

  1. Methods:
    1. 8g/kg of urethane is quite high. 1.5g/kg is considered lethal in rats (Field, et.al. Laboratory Animals, 27: 258-269, 1993).  The standard dose is 1.2g/kg and sometimes used as low as 1.0g.kg in mice.  Can the authors explain why they used such as large dose?
    2. For DSM contractility studies, the authors normalize the maximal response values to bladder weight, however they later show that SCD mice have significant bladder edema, which would increase weight disproportionately to muscle mass and could result in underestimating contractile force in SCD mice. The standard technique for standardization is by cross-sectional area of smooth muscle, measured by histology by fixing the tissue strip after the experiment, sectioning it and staining with H&E.
  2. In the description of Figure 2 in the text, the authors state that bladder compliance was reduced in SCD mice, but that data is missing in the figure.
  3. Additionally, since the authors are concluding that there are contractile deficiencies in SCD mice DSM, it would be nice to see data on residual volume left in the bladder following a void (or a measure of voided volume compared to the calculated amount of saline infused).
  4. Also, if there is a deficit in contractile force, as shown in the DSM strip studies, why is there no concomitant reduction in peak micturition pressure in SCD mice in cystometry?
  5. The authors demonstrate that NOX-2 mRNA is increased in SCD mice, but this is not a direct measurement of oxidative stress. The authors should measure oxidative stress by measuring downstream products of oxidative stress, such as lipid peroxidation or DNA adducts like 8-oxodG.  On a related note, the authors attempt to do this by performing a western blot to measure 3-NT expression in SCD bladders.  However, the authors only show one protein band in their blots; oxidative stress should result in increased tyrosine nitration of multiple proteins, resulting in many visible bands on the blot.  The data presented suggests that the authors are only examining the nitration of one particular protein, but don’t reveal what that protein is.  Additionally, β-actin is a protein that is nitrated in response to oxidative stress (Yoon, et. al., Cell Proliferation 43: 584-93, 2010), but the presented data does not show this.  Finally, the antibody used by the authors is not appropriate for use in western blot, as per the manufacturer’s website.
  6. MPO activity is often associated with bacterial infection, and SCD patients are more susceptible to UTIs (Bruno, et. al., J Urology 166:803-11, 2001). Could the results presented by the authors be a result of bladder infections in the SCD mice?
  7. The authors point out in their conclusions that many of the changes that they observe in their SCD mouse model are similar to those observed in rodent models of bladder outlet obstruction. The authors do not examine urethral function in their SCD mice, but could not the effects of SCD be mediated by urethral dysfunction?
  8. Finally, all of the data presented is correlative, there is no data presented that conclusively demonstrates that oxidative stress is necessary for the pathophysiological changes in bladder function? For example, could the changes be reversed if NOX-2 were inhibited or knocked out?  Could antioxidant treatment reverse the changes? 

Author Response

We thank the reviewer for the thoughtful and well-articulated comments. We have carefully addressed each point raised.

Major Concerns:

  1. Methods:
    1. 8g/kg of urethane is quite high. 1.5g/kg is considered lethal in rats (Field, et.al. Laboratory Animals, 27: 258-269, 1993).  The standard dose is 1.2g/kg and sometimes used as low as 1.0g.kg in mice.  Can the authors explain why they used such as large dose?
  • Response: The dose of 1.8 g/kg i.p. used in our study follows established protocols for mice, not rats, and is commonly adopted to ensure deep and stable anesthesia during prolonged experimental procedures, particularly in urodynamic assessments where movement artifacts and incomplete anesthesia may compromise data reliability. This urethane dose has also been used in previous studies involving similar experimental protocols (Karakus et al., Am J Physiol Renal Physiol., 2019; 317(3):F540–F546; de Oliveira MG et al., Am J Physiol Renal Physiol., 2018; 315(3):F460–F468).
  • 2. For DSM contractility studies, the authors normalize the maximal response values to bladder weight, however they later show that SCD mice have significant bladder edema, which would increase weight disproportionately to muscle mass and could result in underestimating contractile force in SCD mice. The standard technique for standardization is by cross-sectional area of smooth muscle, measured by histology by fixing the tissue strip after the experiment, sectioning it and staining with H&E.
  • Response: We appreciate the reviewer’s insightful comment regarding the potential influence of bladder edema on contractility normalization. Indeed, we observed an increase in total bladder weight in SCD mice, consistent with tissue edema. However, we would like to clarify that for the DSM contractility experiments, the normalization was performed using the weight of the isolated detrusor strips, not the whole bladder. Importantly, the average weight of the strips used for functional recordings did not differ significantly between the control and SCD groups (4.8 ± 0.49 mg vs. 4.45 ± 0.27 mg, respectively, n=16).

2. In the description of Figure 2 in the text, the authors state that bladder compliance was reduced in SCD mice, but that data is missing in the figure.

Response: In the revised version of the manuscript, we have now included bladder compliance data in Figure 2C. The figure legend and corresponding description in the Results section have been updated accordingly. Figure 2 now presents: (A) Frequency of voiding, (B) frequency of NVCs, (C) compliance (D) peak pressure, (E) pressure threshold and (F) capacity.

3. Additionally, since the authors are concluding that there are contractile deficiencies in SCD mice DSM, it would be nice to see data on residual volume left in the bladder following a void (or a measure of voided volume compared to the calculated amount of saline infused).

Response: We agree that the assessment of residual volume or voiding efficiency would provide important complementary information regarding detrusor contractile function in vivo. In this study, our in vivo cystometric setup did not include direct collection of voided or residual volumes.

4. Also, if there is a deficit in contractile force, as shown in the DSM strip studies, why is there no concomitant reduction in peak micturition pressure in SCD mice in cystometry?

Response: We appreciate the reviewer’s thoughtful comment. Indeed, the detrusor smooth muscle from SCD mice exhibited reduced contractile responses in isolated strip experiments. However, peak micturition pressure in vivo is influenced by multiple physiological factors beyond smooth muscle contractility, including neural control, bladder outlet resistance, and timing of voiding reflexes.

In our cystometric protocol, animals were anesthetized and the bladder was catheterized via a suprapubic approach, with minimal influence from abdominal pressure due to anesthesia and surgical exposure. However, compensatory mechanisms may be activated in vivo, such as increased cholinergic drive, altered afferent signaling, or changes in urethral dynamics, which could transiently maintain or even augment peak voiding pressure despite intrinsic detrusor dysfunction.

Therefore, we believe that the reduction in DSM contractility observed ex vivo reflects a true muscular deficit, whereas the cystometric parameters capture a more integrated response, which may partially compensate for or obscure the isolated impairment.

5. The authors demonstrate that NOX-2 mRNA is increased in SCD mice, but this is not a direct measurement of oxidative stress. The authors should measure oxidative stress by measuring downstream products of oxidative stress, such as lipid peroxidation or DNA adducts like 8-oxodG.  On a related note, the authors attempt to do this by performing a western blot to measure 3-NT expression in SCD bladders.  However, the authors only show one protein band in their blots; oxidative stress should result in increased tyrosine nitration of multiple proteins, resulting in many visible bands on the blot.  The data presented suggests that the authors are only examining the nitration of one particular protein, but don’t reveal what that protein is.  Additionally, β-actin is a protein that is nitrated in response to oxidative stress (Yoon, et. al., Cell Proliferation 43: 584-93, 2010), but the presented data does not show this.  Finally, the antibody used by the authors is not appropriate for use in western blot, as per the manufacturer’s website.

Response: We thank the reviewer for this detailed and constructive comment. In our Western blot analysis, we consistently observed two distinct bands at approximately 50 kDa and 25 kDa across biological replicates. Although we did not perform mass spectrometry or co-immunoprecipitation to identify these proteins, similar banding patterns have been reported in the literature, including studies using 3-nitrotyrosine antibodies in mouse penile tissue (Lagoda et al., FASEB J. 2014 Jan;28(1):76-84). Additionally, customer-submitted Western blot images on the Abcam website using the same antibody (ab7048) show comparable bands at ~50 and ~25 kDa in mouse aorta, liver, and lung, suggesting that these molecular weights may correspond to proteins commonly nitrated under oxidative stress conditions across multiple tissues.

We acknowledge that the identity of the nitrated proteins in our samples remains to be determined, and we now clearly state this limitation in the revised Discussion section. We also agree that the inclusion of complementary markers of oxidative damage (such as lipid peroxidation or DNA oxidation products like 8-oxodG) would strengthen our findings. Unfortunately, sample availability limited our ability to perform additional assays in the current study, but we will prioritize these approaches in future experiments.

Regarding the antibody used, we recognize that the manufacturer (Abcam) does not explicitly list Western blot as a validated application for clone HM11 (ab7048) on the product datasheet. However, this antibody has been widely used for Western blot in peer-reviewed publications and in user-submitted experimental data on platforms such as Abcam and CiteAb (https://www.citeab.com/antibodies/769582-ab7048-anti-nitrotyrosine-antibody-hm-11). Based on this accumulated evidence and prior optimization in our laboratory, we selected this antibody for its recognized utility in detecting tyrosine-nitrated proteins under oxidative stress conditions.

6. MPO activity is often associated with bacterial infection, and SCD patients are more susceptible to UTIs (Bruno, et. al., J Urology 166:803-11, 2001). Could the results presented by the authors be a result of bladder infections in the SCD mice?

Response: In our study, all animals were housed under specific pathogen-free (SPF) conditions, and no overt signs of infection (e.g., hematuria, pyuria, or abnormal urine appearance) were observed during handling or bladder catheterization. Moreover, the elevated MPO activity observed in SCD mice was accompanied by histological evidence of inflammatory cell infiltration and increased mRNA expression of IL-1β in bladder tissue, supporting the presence of a sterile inflammatory response.

7. The authors point out in their conclusions that many of the changes that they observe in their SCD mouse model are similar to those observed in rodent models of bladder outlet obstruction. The authors do not examine urethral function in their SCD mice, but could not the effects of SCD be mediated by urethral dysfunction?

Response: We thank the reviewer for this insightful comment. It is true that urethral dysfunction can contribute to bladder remodeling and dysfunction, and we did not directly assess urethral function in this study. While our findings of increased oxidative stress, inflammation, altered detrusor contractility, and impaired compliance support a primary bladder pathology, we cannot exclude the possibility that some of these changes may be secondary to functional urethral obstruction or altered outlet resistance in SCD mice.

8. Finally, all of the data presented is correlative, there is no data presented that conclusively demonstrates that oxidative stress is necessary for the pathophysiological changes in bladder function? For example, could the changes be reversed if NOX-2 were inhibited or knocked out?  Could antioxidant treatment reverse the changes? 

Response: It is important to acknowledge that the present findings are correlative and do not establish a direct causal relationship between oxidative stress and bladder dysfunction in SCD mice. Further studies using pharmacological inhibition or genetic deletion of NOX-2, as well as antioxidant interventions, are needed to determine whether oxidative stress plays a mechanistic role in the observed alterations in bladder structure and function.

Reviewer 2 Report

Comments and Suggestions for Authors

This article is relatively reasonable and reliable. The aim of this study was to investigate the relationship between oxidative stress, inflammation, and bladder dysfunction in the Townes transgenic SCD mouse model. They found that evidence indicating that CD mice exhibit an OAB phenotype, characterized by increased frequencies of voiding and non-voiding contrac- tions and reduced bladder compliance. These changes suggest that oxida-tive and nitrosative stress, combined with inflammation, contribute to bladder remodeling and dysfunction in SCD. It is a topic of interest to the researchers in the related areas and This is the first study to characterize bladder alterations in Townes SCD mice, establishing this model as a valuable tool for investigating lower urinary tract complications in SCD, but the article has severe problems.

1.The methods of cultivating Townes transgenic sickle cell mice should be provided.

  1. The images of histology didn’t have scales, please supplement.

3.The in vivo test need add positive drug group and negative drug group.

4.All figures need to redesign.   

Author Response

We thank the reviewer for the overall assessment and constructive comments. Please find our point-by-point responses below:

1.The methods of cultivating Townes transgenic sickle cell mice should be provided.

Response: This information has been included in the Materials and Methods section. Breeding pairs of Townes transgenic sickle cell mice were originally developed at The Jackson Laboratory and acquired by CEMIB-UNICAMP, where local breeding and genotyping were performed to identify homozygous animals. Experimentally eligible mice were then transferred to the Hemocentro-UNICAMP facility, where all experimental procedures were conducted.

2. The images of histology didn’t have scales, please supplement.

Response: We thank the reviewer for the observation. We would like to clarify that all histological images already contain scale bars; however, we acknowledge that they may have been difficult to visualize due to their small size. In response, we have increased the size and contrast of the scale bars in the revised figure to improve clarity. The corresponding units (200 µm in panels A and D; 50 µm in panels B, C, E, and F) are also specified in the updated figure legend.

3. The in vivo test need add positive drug group and negative drug group.

Response: We acknowledge that including pharmacological intervention groups would strengthen the interpretation of mechanistic pathways. However, the current study was designed as an observational investigation aimed at characterizing the baseline bladder phenotype in Townes SCD mice. The inclusion of positive or negative drug controls was beyond the scope of this work. Nonetheless, future studies are planned to incorporate antioxidant or NOX-2 inhibitor treatments to assess causality and therapeutic potential.

4.All figures need to redesign.   

Response: Figures have been revised for improved clarity and consistency. Font size, resolution, labeling, and layout have been adjusted to enhance readability. We hope the updated versions meet the journal’s standards and the reviewer’s expectations.

Round 2

Reviewer 1 Report

Comments and Suggestions for Authors

The authors have adequately addressed my earlier concerns.

Author Response

Comments 1: The authors have adequately addressed my earlier concerns.
Response 1: We thank the reviewer for the positive feedback and are pleased to know that our revisions have satisfactorily addressed the previous concerns.